# MINE: MUTUAL INFORMATION NEURAL ESTIMATION

## ABSTRACT

We argue that the estimation of the mutual information between high dimensional continuous random variables is achievable by gradient descent over neural networks. This paper presents a Mutual Information Neural Estimator (MINE) that is linearly scalable in dimensionality as well as in sample size. MINE is back-propable and we prove that it is strongly consistent. We illustrate a handful of applications in which MINE is succesfully applied to enhance the property of generative models in both unsupervised and supervised settings. We apply our framework to estimate the information bottleneck, and apply it in tasks related to supervised classification problems. Our results demonstrate substantial added flexibility and improvement in these settings.

## 1 INTRODUCTION

Mutual information is an important quantity for expressing and understanding the relationship between random variables. As a fundamental tool of data science, it has found application in a range of domains and tasks, including applications to biomedical sciences, blind source separation (BSS, e.g., independent component analysis, Hyvärinen et al., 2004), information bottleneck (IB, Tishby et al., 2000), feature selection (Kwak & Choi, 2002; Peng et al., 2005), and causality (Butte & Kohane, 2000).

In contrast to correlation, mutual information captures the absolute statistical dependency between two variables, and thus can act as a measure of true dependence. Put simply, mutual information is the shared information of two random variables, $X$ and $Z$, defined on the same probability space, $(\mathcal{X} \times \mathcal{Z}, \mathcal{F})$, where $\mathcal{X} \times \mathcal{Z}$ is the domain over both variables (such as $\mathbb{R}^m \times \mathbb{R}^n$), and $\mathcal{F}$ is the set of all possible outcomes over both variables. The mutual information has the form [1]:

$$I(X; Z) = \int_{\mathcal{X} \times \mathcal{Z}} \log \frac{d\mathbb{P}_{XZ}}{d\mathbb{P}_X \otimes \mathbb{P}_Z} d\mathbb{P}_{XZ} \tag{1}$$

where $\mathbb{P}_{XZ} : \mathcal{F} \to [0, 1]$ is a probabilistic measure (commonly known as a joint probability distribution in this context), and $\mathbb{P}_X = \int_{\mathcal{Z}} d\mathbb{P}_{XZ}$ and $\mathbb{P}_Z = \int_{\mathcal{X}} d\mathbb{P}_{XZ}$ are the marginals.

The mutual information is notoriously difficult to compute. Exact computation is only tractable with discrete variables (as the sum can be computed exactly) or with a limited family of problems where the probability distributions are known and for low dimensions. For more general problems, common approaches include binning (Fraser & Swinney, 1986; Darbellay & Vajda, 1999), kernel density estimation (Moon et al., 1995; Kwak & Choi, 2002), Edgeworth expansion based estimators Van Hulle (2005) and likelihood-ratio estimators based on support vector machines (SVMs, e.g., Suzuki et al., 2008). While the mutual information can be estimated from empirical samples with these estimators, they still make critical assumptions about the underlying distribution of samples, and estimate errors can reflect this. In addition, these estimators typically do not scale well with sample size or dimension.

More recently, there has been great progress in the estimation of $f$-divergences (Nguyen et al., 2010) and integral probability metrics (IPMs, Sriperumbudur et al., 2009) using deep neural networks (e.g., in the context of $f$-divergences and the Wasserstein distance or Fisher IPMs, Nowozin et al., 2016; Arjovsky et al., 2017; Mroueh & Sercu, 2017). These methods are at the center of generative adversarial networks (GANs Goodfellow et al., 2014), which train a generative model without any explicit

---

[1]We assume the convention that $\log$ is the natural log, so that our units of information are in *nats*.

assumptions about the underlying distribution of the data. One perspective on these works is that, given the correct constraints on a neural network, the network can be used to compute a variational lower-bound on the distance or divergence of implicit probability measures.

In this paper we look to extend this estimation strategy to mutual information as given in equation 1, which we note corresponds to the Kullback-Leibler (KL-) divergence Kullback (1997) between the joint, $\mathbb{P}_{XZ}$ and the product of the marginal distributions, $\mathbb{P}_X \otimes \mathbb{P}_Z$, i.e., $D_{KL}(\mathbb{P}_{XZ} \,||\, \mathbb{P}_X \otimes \mathbb{P}_Z)$. This observation can be used to help formulate variational Bayes in terms of implicit distributions (Mescheder et al., 2017) or INFOMAX (Brakel & Bengio, 2017).

We introduce an estimator for the mutual information based on the Donsker-Varadhan representation of the KL-divergence (Ruderman et al., 2012). As with those introduced by Nowozin et al. (2016), our estimator is scalable, flexible, and is completely trainable via back-propagation. The contributions of this paper are as follows.

- We introduce the mutual information neural estimator (MINE), providing its theoretical bases and generalizability to other information metrics.
- We illustrate that our estimator can be used to train a model with improved support coverage and richer learned representation for training adversarial models (such as adversarially-learned inferences, ALI, Dumoulin et al., 2016).
- We demonstrate how to use MINE to improve reconstructions and inference in Adversarially Learned Inference Dumoulin et al. (2016) on large scale Datasets.
- We show that our estimator provides a method of performing the Information Bottleneck method Tishby et al. (2000) in a continuous setting, and that this approach outperforms variational bottleneck methods (Alemi et al., 2016).

## 2 BACKGROUND

### 2.1 MUTUAL INFORMATION

Mutual information is a Shannon entropy-based measure of dependence between random variables. Following the definition in Equation 1, the mutual information can be understood as the decrease in the uncertainty of $X$ given $Z$:

$$I(X; Z) := H(X) - H(X \mid Z) = H(Z) - H(Z \mid X), \tag{2}$$

where $H$ is the Shannon entropy and $H(Z \mid X)$ is the conditional entropy of $Z$ given $X$ (the amount of information in $Z$ not given from $X$). Using simple manipulation, we write the mutual information as the Kullback-Leibler (KL-) divergence between the joint, $\mathbb{P}_{XZ}$, and the product of the marginals $\mathbb{P}_X \otimes \mathbb{P}_Z$:

$$I(X; Z) = H(X) + H(Z) - H(X, Z) = D_{KL}(\mathbb{P}_{XZ} \,||\, \mathbb{P}_X \otimes \mathbb{P}_Z), \tag{3}$$

where $H(X, Z)$ is the joint entropy of $X$ and $Z$. It can be noted here that the mutual information is zero exactly when the KL-divergence is zero. The intuitive meaning is immediately clear: the larger the divergence between the joint and the product of the marginals, the stronger the dependence between $X$ and $Z$.

There is also a strong connection between the mutual information and the structure between random variables. We briefly touch upon this subject in Appendix 6.1.

### 2.2 THE DONSKER-VARADHAN BOUND

MINE relies on the Donsker-Varadhan representation of the KL-divergence, which provides a tight lower-bound on the mutual information. The KL-divergence between two probability distributions $\mathbb{P}$ and $\mathbb{Q}$ on a measure space $\Omega$, with $\mathbb{P}$ absolutely continuous with respect to $\mathbb{Q}$, is defined as

$$D_{KL}(\mathbb{P} \,||\, \mathbb{Q}) := \int_\Omega \log\left(\frac{d\mathbb{P}}{d\mathbb{Q}}\right) d\mathbb{P} = \mathbb{E}_\mathbb{P}\left[\log \frac{d\mathbb{P}}{d\mathbb{Q}}\right] \tag{4}$$

where the argument of the log is the density ratio[2] and $\mathbb{E}_{\mathbb{P}}$ denotes the expectation with respect to $\mathbb{P}$. It follows from Jensen's inequality that the KL-divergence is always non-negative and vanishes if and only if $\mathbb{P} = \mathbb{Q}$.

The following theorem gives a variational representation of the KL-divergence:

**Theorem 1** (Donsker-Varadhan representation). *The KL divergence between any two distributions $\mathbb{P}$ and $\mathbb{Q}$, with $\mathbb{P} \ll \mathbb{Q}$, admits the following dual representation (Donsker & Varadhan, 1983):*

$$D_{KL}(\mathbb{P} \,||\, \mathbb{Q}) = \sup_{T:\Omega \to \mathbb{R}} \mathbb{E}_{\mathbb{P}}[T] - \log(\mathbb{E}_{\mathbb{Q}}[e^T]) \tag{5}$$

*where the supremum is taken over all functions $T$ such that the two expectations are finite. Given any subclass $\mathcal{F}$ of such functions, this yields the lower bound:*

$$D_{KL}(\mathbb{P} \,||\, \mathbb{Q}) \geq \sup_{T \in \mathcal{F}} \mathbb{E}_{\mathbb{P}}[T] - \log(\mathbb{E}_{\mathbb{Q}}[e^T]) \tag{6}$$

The bound in Equation 6 is known as the *compression lemma* in the PAC-Bayes literature (Banerjee, 2006). A simple proof goes as follows. Given $T \in \mathcal{F}$, consider the Gibbs distribution $\mathbb{G}$ defined by $d\mathbb{G} = \frac{1}{Z}e^T d\mathbb{Q}$, where $Z = \mathbb{E}_{\mathbb{Q}}[e^T]$. By construction,

$$\mathbb{E}_{\mathbb{P}}[T] - \log Z = \mathbb{E}_{\mathbb{P}}\left[\log \frac{d\mathbb{G}}{d\mathbb{Q}}\right] \tag{7}$$

The gap $\Delta$ between left and right hand sides of Equation 6 can then be written as:

$$\Delta = \mathbb{E}_{\mathbb{P}}\left[\log \frac{d\mathbb{P}}{d\mathbb{Q}} - \log \frac{d\mathbb{G}}{d\mathbb{Q}}\right] = \mathbb{E}_{\mathbb{P}} \log \frac{d\mathbb{P}}{d\mathbb{G}} = D_{KL}(\mathbb{P} \,||\, \mathbb{G}) \geq 0 \tag{8}$$

and we conclude by the positivity of the KL-divergence. The identity (8) also shows that the bound is tight whenever $\mathbb{G} = \mathbb{P}$, namely for optimal functions $T^*$ taking the form

$$T^* = \log \frac{d\mathbb{P}}{d\mathbb{Q}} + C \tag{9}$$

for some constant $C \in \mathbb{R}$.

It is interesting to compare the Donsker-Varadhan bound with other variational bounds proposed in the literature. The variational divergence estimation proposed in (Nguyen et al., 2010) and used in Nowozin et al. (2016) and Mescheder et al. (2017), leads to the following bound:

$$D_{KL}(\mathbb{P} \,||\, \mathbb{Q}) \geq \sup_{T \in \mathcal{F}} \mathbb{E}_{\mathbb{P}}[T] - \mathbb{E}_{\mathbb{Q}}[e^{T-1}] \tag{10}$$

Although both bounds are tight for sufficiently large families $\mathcal{F}$, the Donsker-Varadhan bound is *stronger* in the sense that for any fixed $T$, the right hand side of Equation 6 is larger than the right hand side[3] of Equation 10. We perform numerical comparisons in Section 4.1.

We refer to the work by Ruderman et al. (2012) for a derivation of both representations (6) and (10) from unifying point of view of Fenchel duality, in the more general context of $f$-divergences.

## 3 THE MUTUAL INFORMATION NEURAL ESTIMATOR

### 3.1 DEFINITION

We are interested in the case of a *joint* random variable $(X, Z)$ on a joint probability space $\Omega = \mathcal{X} \times \mathcal{Z}$, and where $\mathbb{P} = \mathbb{P}_{XZ}$ is the joint distribution, $\mathbb{Q} = \mathbb{P}_X \otimes \mathbb{P}_Z$ is the product distribution. $\mathbb{P}$ is then absolutely continuous with respect to $\mathbb{Q}$. Using the expression (3) for the mutual information in terms of a KL-divergence, we obtain the following representation:

$$I(X; Z) \geq \sup_{T \in \mathcal{F}} \mathbb{E}_{\mathbb{P}_{XZ}}[T(x, z)] - \log(\mathbb{E}_{\mathbb{P}_X \otimes \mathbb{P}_Z}[e^{T(x,z)}]). \tag{11}$$

---

[2]Although the discussion is more general, we can think of $\mathbb{P}$ and $\mathbb{Q}$ as being distributions on some compact domain $\Omega \subset \mathbb{R}^d$, with density $p$ and $q$ respect the Lebesgue measure $\lambda$, so that $D_{KL} = \int p \log \frac{p}{q} d\lambda$.

[3]To see this, just apply the identity $x \geq e \log x$ with $x = \mathbb{E}_{\mathbb{Q}}[e^T]$.

The inequality in Equation 11 is intuitive in terms of deep learning optimization. The idea is to parametrize the functions $T : \mathcal{X} \times \mathcal{Z} \to \mathbb{R}$ in $\mathcal{F}$ by a deep neural network with parameters $\theta \in \Theta$, turning the infinite dimensional problem into a much easier parametric optimization problem. In the following we call $T_\theta$ the *statistic network*. The expectations in the above lower-bound can then be estimated by Monte-Carlo (MC) sampling using empirical samples $(x, z) \sim \mathbb{P}_{XZ}$. Samples $\bar{x} \sim \mathbb{P}_X$ and $\bar{z} \sim \mathbb{P}_Z$ from the marginals are obtained by simply dropping $x, z$ from samples $(\bar{x}, z)$ and $(x, \bar{z}) \sim \mathbb{P}_{XZ}$. The objective can be maximized by gradient ascent.

In what follows we use the notation $\hat{\mathbb{P}}_X^{(n)}$ for the empirical distribution associated to a given set of $n$ *iid* samples drawn for $\mathbb{P}_X$. If we denote

$$\hat{\theta}_n = \arg\sup_{\theta \in \Theta} \mathbb{E}_{\hat{\mathbb{P}}_{XZ}^{(n)}}[T_\theta(x, z)] - \log(\mathbb{E}_{\hat{\mathbb{P}}_X^{(n)} \otimes \hat{\mathbb{P}}_Z^{(n)}}[e^{T_\theta(x,z)}]) \tag{12}$$

as the optimal set of parameters under the above conditions, we obtain the *Mutual Information Neural Estimator* (MINE):

**Definition 3.1** (Mutual information neural estimator (MINE))**.**

$$\widehat{I(X; Z)}_n = \mathbb{E}_{\hat{\mathbb{P}}_{XZ}^{(n)}}[T_{\hat{\theta}_n}(x, z)] - \log(\mathbb{E}_{\hat{\mathbb{P}}_X^{(n)} \otimes \hat{\mathbb{P}}_Z^{(n)}}[e^{T_{\hat{\theta}_n}(x,z)}]). \tag{13}$$

Algorithm 1 presents details of the implementation of MINE.

---

**Algorithm 1** . Mutual Information Estimation

$\theta \leftarrow$ initialize network parameters
**repeat**
$\quad (x^{(1)}, z^{(1)}), \ldots, (x^{(n)}, z^{(n)}) \sim \mathbb{P}_{XZ} \qquad\qquad$ ▷ Draw $n$ samples from the joint distribution
$\quad \bar{z}^{(1)}, \ldots, \bar{z}^{(n)} \sim \mathbb{P}_Z \qquad\qquad\qquad\quad$ ▷ Draw $n$ samples from the $Z$ marginal distribution
$\quad \mathcal{V}(\theta) \leftarrow \frac{1}{n} \sum_{i=1}^n T_\theta(x^{(i)}, z^{(i)}) - \log(\frac{1}{n} \sum_{i=1}^n e^{T_\theta(x^{(i)}, \bar{z}^{(i)})})$
$\qquad\qquad\qquad\qquad\qquad\qquad\qquad\qquad\qquad\quad$ ▷ Evaluate the lower-bound
$\quad \theta \leftarrow \theta + \nabla_\theta \mathcal{V}(\theta) \qquad\qquad\qquad\quad$ ▷ Update the statistic network parameters
**until** convergence

---

We will also use an adaptive gradient clipping method to ensure stability whenever MINE is used in conjunction with another adversarial objective. The details of this are provided in Appendix 6.3.

## 3.2 Consistency

In this section we discuss the *consistency* of MINE. The estimator relies on $(i)$ a neural network architecture and $(ii)$ a choice of $n$ samples from the data distribution $\mathbb{P}_{XZ}$. We define consistency in the following way:

**Definition 3.2** (Consistency)**.** The estimator $\widehat{I(X; Z)}_n$ is (strongly) consistent if for all $\epsilon > 0$, then there exists a positive integer $N$ and a choice of neural network architecture such that:

$$\forall n \geq N, \quad |I(X, Z) - \widehat{I(X; Z)}_n| \leq \epsilon \text{ with probability one}$$

In other words, the estimator converges to the true mutual information as $n \to \infty$, almost surely over the choice of samples. The question of consistency breaks into two problems: an *approximation* problem related to the size of the family $\mathcal{F}$, and inducing the gap in the inequality (11) ; and an *estimation* problem related to the use of empirical measures in (12). The first problem is addressed by the universal approximation theorem for neural networks (Hornik, 1989). For the second problem, classical consistency theorems for extremum estimators apply (Van de Geer, 2000), under mild conditions on the parameter space.

This leads to the two lemmas below. The proofs are given in Appendix 6.2. In what follows we use the notation $\hat{I}[T]$ for the argument of the supremum in Equation (11):

$$\hat{I}[T] := \mathbb{E}_{\mathbb{P}_{XZ}}[T] - \log(\mathbb{E}_{\mathbb{P}_X \otimes \mathbb{P}_Z}[e^T])$$

**Lemma 1.** *Let $\eta > 0$. There exists a feedforward network function $T_{\hat{\theta}} \colon \Omega \to \mathbb{R}$ such that*

$$|I(X, Z) - \hat{I}(T_{\hat{\theta}})| \leq \eta$$

*A fortiori if $\mathcal{F}$ is any family of functions having $T_{\hat{\theta}}$ as one of its elements,*

$$|I(X, Z) - \sup_{T_\theta \in \mathcal{F}} \hat{I}(T_\theta)| \leq \eta \tag{14}$$

**Lemma 2.** *Let $\eta > 0$. Let $\mathcal{F}$ be the family of functions $T_\theta \colon \Omega \to \mathbb{R}$ defined by a given network architecture. We assume the parameters $\theta$ are restricted to some compact domain $\Theta \subset \mathbb{R}^k$. Then there exists $N \in \mathbb{N}$ such that*

$$\forall n \geq N, \quad |\widehat{I(X; Z)}_n - \sup_{T_\theta \in \mathcal{F}} \hat{I}(T_\theta)| \leq \eta \ \text{with probability one} \tag{15}$$

These results lead to the following consistency theorem.

**Theorem 2.** *MINE as defined by Equ. 12 and 13 is a (strongly) consistent.*

*Proof.* Let $\epsilon > 0$. We apply the two Lemma to find a a family of neural network function $\mathcal{F}$ and $N \in \mathbb{N}$ such that (15) and (14) hold with $\eta = \epsilon/2$. By the triangular inequality, for all $n \geq N$ and with probability one, we have that

$$|I(X, Z) - \widehat{I(X; Z)}_n| \leq |I(X, Z) - \sup_{T_\theta \in \mathcal{F}} \hat{I}(T_\theta)| + |\widehat{I(X; Z)}_n - \sup_{T_\theta \in \mathcal{F}} \hat{I}(T_\theta)| \leq \epsilon \tag{16}$$

which proves consistency. □

### 3.3 GENERALIZATION TO $f$-INFORMATION MEASURES

We close this section by pointing out that the previous construction can be extended to more general information measures based on so-called $f$-divergences (Ali & Silvey, 1966):

$$D_f(\mathbb{P} \mid\mid \mathbb{Q}) := \int_\Omega f\left(\frac{d\mathbb{P}}{d\mathbb{Q}}\right) d\mathbb{Q} \tag{17}$$

indexed by a convex function $f \colon [0, \infty) \to \mathbb{R}$ such that $f(1) = 0$. The KL-divergence is a special case of $f$-divergence with $f(u) = u \log(u)$. Just as the mutual information can be understood as the KL-divergence between the joint and product of marginals distributions, we can define a family of $f$-information measures as $f$-divergences:

$$I_f(X; Z) := D_f(\mathbb{P}_{XZ} \mid\mid \mathbb{P}_X \otimes \mathbb{P}_Z) \tag{18}$$

An analogue for $f$-divergences of the Donsker-Varadhan representation of Theorem 1 can be found in Ruderman et al. (2012). The key idea is to express $f$-divergences in terms of convex operators, and to leverage Fenchel-Legendre duality to obtain variational representation in terms of the convex conjugate (Rockafellar, 1970). This allows a straightforward extension of MINE to a mutual $f$-information estimator, following the construction of of the previous section. The study of such information measures and their estimators is left for future work.

## 4 APPLICATIONS AND EXPERIMENTS

In this section, we present applications of mutual information through the mutual information neural estimator (MINE), as well as competing methods that are designed to achieve the same goals. We also present experimental results touching on each of these applications.

### 4.1 MUTUAL INFORMATION ESTIMATION

Mutual information is an important quantity for analyzing and understanding the statistical dependencies between random variables. The most straightforward application for MINE then is estimation of the mutual information.

**Related works on estimating mutual information**    There are a number of methods that can also be used to estimate mutual information given only empirical samples of the joint distribution of variables of interest. The fundamental difficulty in estimation is the intractability of joint and product of marginals, as exact computation requires integration over the joint of continuous variables. Kraskov et al. (2004) proposes a $k$-NN estimator based on estimating the entropy terms of the mutual information; and this comes with the usual limitations of non-parametric methods. Van Hulle (2005) presents an estimator built around the Edgeworth series (Hall, 2013). The entropy of the distribution is approximated by a Gaussian with additional correction brought by higher-order cumulants. This method is only tractable in very low-dimensional data and breaks down when departure from Gaussianity is too severe. Suzuki et al. (2008) exploits a likelihood-ratio estimator using kernel methods. Other recent works include Kandasamy et al. (2017); Singh & Pczos (2016); Moon et al. (2017).

MINE, on the other hand, inherits all the benefits of neural networks in scalability and can, in principle, calculate the mutual information using a large number of high-dimensional samples. We posit then that, given empirical samples of two random variables, $X$ and $Z$, and a high-enough capacity neural network, MINE will provide good estimates for the mutual information without the necessary constraints of the methods mentioned above.

**Experiment: estimating mutual information between two Gaussians**    We begin by comparing MINE to the $k$-means-based non-parametric estimator found in Kraskov et al. (2004). In our experiment, we consider two bivariate Gaussian random variables $X_a$ and $X_b$ with correlation, $corr(X_a, X_b) = \rho \in [-0.99, -0.9, -0.7, -0.5, -0.3, -0.1, 0., 0.1, 0.3, 0.5, 0.7, 0.9, 0.99]$. As the mutual information is invariant to continuous bijective transformation of the considered variables, it is enough to consider standardized Gaussians marginals. We also compare two versions of MINE: the version of the current paper based on the Donsker-Varadhan representation 5 of the KL divergence; and the one based on the $f$-divergence representation 10 proposed by Nguyen et al. (2010) and used in Nowozin et al. (2016) and Mescheder et al. (2017).

Our results are presented in Figure 1 and 2. We observe that both MINE and Kraskov's estimation are virtually indistinguishable from the ground truth; and that MINE provides a much tighter estimate of the mutual information than the version using the bound of Nguyen et al. (2010).

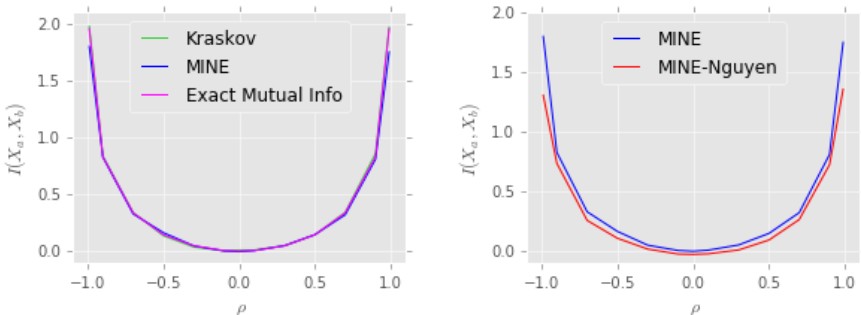

Figure 1: Mutual information between two bivariate Gaussians with component-wise correlation of $corr(X_a, X_b) = \rho \in [-0.99, -0.9, -0.7, -0.5, -0.3, -0.1, 0., 0.1, 0.3, 0.5, 0.7, 0.9, 0.99]$.

## 4.2    ENTROPY MAXIMIZED GANS TO IMPROVE GENERATIVE SUPPORT

Mode-dropping (Che et al., 2016) is a common pathology of generative adversarial networks (GANs, Goodfellow et al., 2014) where the generator does not generate all of the modes in the target dataset (such as not generating images that correspond to specific labels). We identify at least two source of mode dropping in GANs:

- **Discriminator liability:** In this case, the discriminator classifies only a fraction of the real data as real. As a consequence of this, there is no gradient for the generator to learn to generate modes that have poor representation under the discriminator.

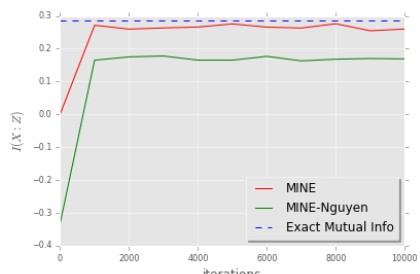
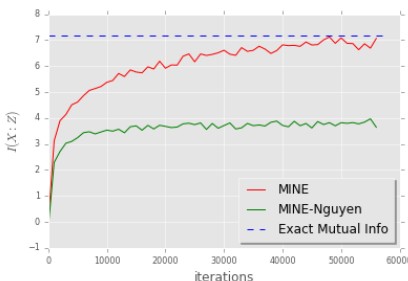

(a) Mutual Information estimation on gaussians of dimension 2

(b) Mutual Information estimation between two Gaussians of dimension 50

Figure 2: Estimates of the mutual information as a function of the number of iterations.

- **Generator liability:** The generator is greedy and concentrates its probability mass on the smallest subset most likely to fool the discriminator. Here, the generator simply focuses on a subset of modes which maximize the discriminator's Bayes risk.

We focus here on the second type of mode-dropping. In order to alleviate the greedy behavior of the generator, we encourage the generator to maximize the entropy of the generated data. This can be achieved by modifying the GAN objective for the generator with a mutual information term.

Our treatment involves the typical GAN setting in Goodfellow et al. (2014). We denote by $p_{\text{real}}$ the real data distribution on $\mathcal{X}$, and by $p_{\text{gen}}$ the generated distribution, induced by a function[4] $G : \mathcal{Z} \to \mathcal{X}$ from a (relatively simple, such as a spherical Gaussian) prior density $p(z)$, so that $\mathbb{E}_{x \sim p_{\text{gen}}}[f(x)] = \mathbb{E}_{z \sim p(z)}[f(G(z)]$ for all functions $f$ on $\mathcal{X}$. In this setting, the discriminator $D : \mathcal{X} \to \mathbb{R}$, which is modeled by a deep neural network with sigmoid nonlinearity, is optimized so as to maximize the value function:

$$V(D, G) = \mathbb{E}_{p_{\text{real}}}[\log D(x)] + \mathbb{E}_{p_{\text{gen}}}[\log (1 - D(x)]. \tag{19}$$

As observed in Nowozin et al. (2016), maximizing the value function amounts to maximizing the variational lower-bound of $2 * D_{JS}(\mathbb{P}||\mathbb{Q}) - 2\log 2$, where $D_{JS}$ is the Jensen-Shannon divergence. The generator is then optimized to minimize $V$ alternatively as the discriminator maximizes it. In practice, however, we will use a *proxy* to be maximized by the generator, $\mathbb{E}_{p_{\text{gen}}}[\log(D(x)]$, which can palliate vanishing gradients.

In order to palliate mode-dropping, our strategy is to maximize the entropy of the generated data. Since $G(Z)$ is a deterministic function of $Z$, the conditional entropy $H(G(Z)|Z)$ is zero and thus

$$I(G(Z); Z) = H(G(Z)) \tag{20}$$

In other words, the entropy can be estimated using MINE. The generator objective then becomes:

$$\arg \max_{G} \mathbb{E}_{p(z)}[\log(D(G(z)))] + \beta I(G(Z); Z). \tag{21}$$

As the samples $G(z)$ are differentiable w.r.t. the parameters of $G$ and MINE is a completely differentiable function, we can maximize the mutual information using back-propagation and gradient descent by only specifying this additional loss term. Since the mutual information is unbounded, we use adaptive gradient clipping to ensure stability (see Appendix 6.3).

**Related works on mode-dropping**  In mode regularized GANs, Che et al. (2016) proposes to learn a reconstruction distribution, then teach the generator to sample from it. The intuition behind this is that the reconstruction distribution is a de-noised or *smoothed* version of the data distribution, and thus easier to learn. However, the connection to reducing mode dropping is only indirect.

InfoGAN (Chen et al., 2016) is a method which attempts to improve mode coverage by leveraging the Agokov and Baber conditional entropy variational lower-bound (Barber & Agakov, 2003). This

---

[4]We can also take a more general probabilistic view where the generator defines a conditional distribution $p(x|z)$. In this setting, the deterministic case corresponds to defining $p(x|z)$ as a Dirac distribution $\delta(x - G(z))$.

bound involves approximating the intractable conditional distribution $\mathbb{P}_{Z|X}$ by using a tractable recognition network, $F : \mathcal{X} \to \mathcal{Z}$. In this setting, the variational approach bounds the *conditional entropy*, $H(X \mid Z)$, which effectively maximizes a variational lower bound on the entropy $H(G(Z))$.

VEEGAN Srivastava et al. (2017), like InfoGAN, makes use of a recognition network to maximize the Agokov and Baber variational lower-bound, but is trained like adversarially learned inference (ALI, Dumoulin et al., 2016, , see the following section for details). Since, at convergence the joint distributions of the generative and recognition networks are matched, this has the effect of minimizing the conditional entropy, $H(X|Z)$.

Our approach is closest to that of Dai et al. (2017), where they also formulated a GAN with entropy regularization of the generator. Interestingly, they show that, in the context of Energy-based GANs, such a regularization strategy yields a discriminator score function that at equilibrium is proportional to the log-density of the empirical distribution. The main difference between their work and our regularized GAN formulation is that we use MINE to estimate entropy while they used a non-parametric estimate that does not scale particularly well with dimensionality of the data domain.

**Experiment: swiss-roll and 25-Gaussians datasets**  Here, we apply MINE to improve mode coverage when training a generative adversarial network (GAN, Goodfellow et al., 2014). Following Equation 21, we estimate the mutual information using MINE and use this estimate to maximize the entropy of the generator. We demonstrate this effect on a Swiss-roll dataset, comparing two models, one with $\beta = 0$ (which corresponds to the orthodox GAN as in Goodfellow et al. (2014)) and one with $\beta = 1.0$, which corresponds to entropy-maximization.

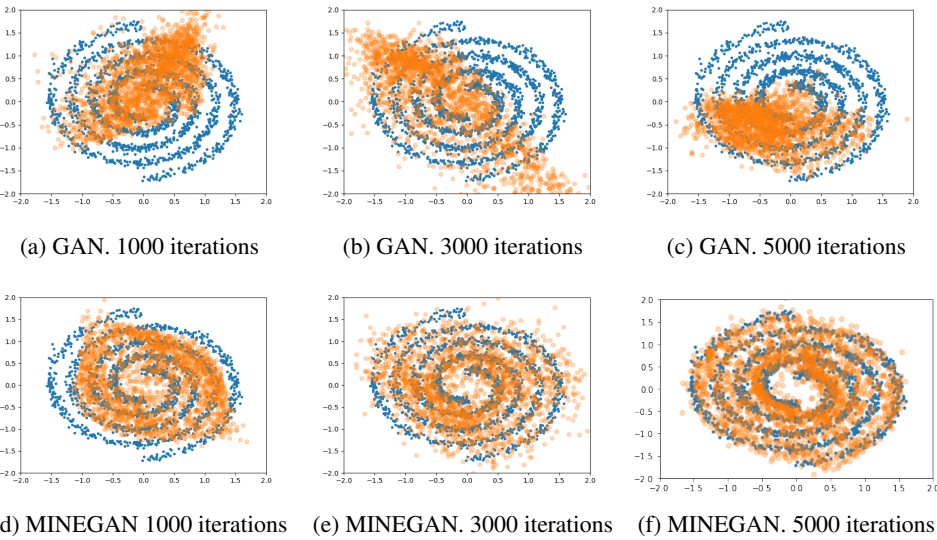

(a) GAN. 1000 iterations     (b) GAN. 3000 iterations     (c) GAN. 5000 iterations

(d) MINEGAN 1000 iterations   (e) MINEGAN. 3000 iterations   (f) MINEGAN. 5000 iterations

Figure 3: The generator of the GAN model without mutual information maximization suffers from mode collapse (has poor coverage of the target dataset). In addition to the the GAN objective, MINEGAN maximizes the mutual information $I(G(Z); Z)$. The MINEGAN generator learns a distribution with a high amount of structured noise. In addition, MINEGAN converges faster, shows better coverage of the ground truth distribution, as well as less mode dropping.

Our results on the swiss-roll (Figure 3) and the 25-Gaussians (Figure 4) datasets show improved mode coverage over the baseline with no mutual information objective. This confirms our hypothesis that maximizing mutual information helps against mode-dropping in this simple setting.

### 4.3 IMPROVING THE REPRESENTATION OF BI-DIRECTIONAL ADVERSARIAL MODELS

Adversarial bi-directional models are an extension of GANs which incorporate a reverse model $F : \mathcal{X} \to \mathcal{Z}$. These were introduced in adversarially-learned inference (ALI, Dumoulin et al., 2016), closely related BiGAN (Donahue et al., 2016), and variants that minimize the condi-

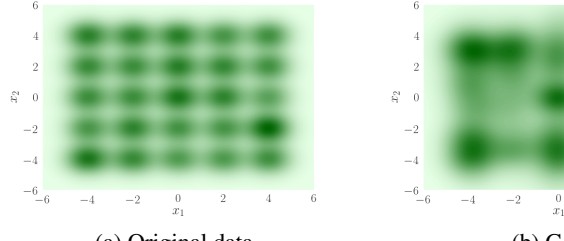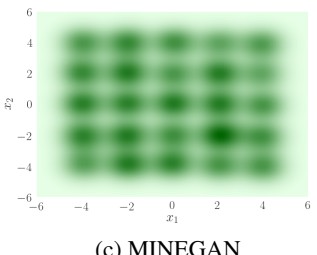

(a) Original data          (b) GAN          (c) MINEGAN

Figure 4: Kernel density estimate (KDE) plots for MINEGAN samples and GAN samples on 25 Gaussians dataset. It is evident from the plot that again MINE does a decent job of capturing all the modes of the distribution while the standard GAN drops quite a few.

tional entropy (ALICE, Li et al., 2017). These models train a discriminator to maximize the value function of Equation 19 over the two joint distributions $p_{\text{enc}}(x, z) = p_{\text{enc}}(z|x)p(x)$ and $p_{\text{dec}}(x, z) = p_{\text{dec}}(x|z)p(z)$ over $\mathcal{X} \times \mathcal{Z}$, induced by the forward (encoder) and reverse (decoder) models, respectively.

In principle, ALI should be able to learn a feature representation as well as palliate mode dropping. However, in practice ALI guarantees neither due to identifiability issues (Li et al., 2017). This is further evident as the generated samples from the forward model can be poor reconstructions of the data given the inferred latent representations from the reverse model. In order to address these issues, ALICE introduces an additional term to minimize the conditional entropy by minimizing the reconstruction error.

To demonstrate the connection to mutual information, it can be shown (see the Appendix, Section 6.4, for a proof) that the reconstruction error is bounded as:

$$\mathcal{R} \leq D_{KL}(p_{\text{enc}} \,||\, p_{\text{dec}}) - I_{p_{\text{enc}}}(X, Z) + H_{p_{\text{enc}}}(Z) \tag{22}$$

If $H_{p_{\text{enc}}}(Z)$ is fixed (which can be accomplished in how the reverse model is defined), then matching the joint distributions during training in addition to maximizing the mutual information between $X$ and $Z$ will lower the reconstruction error.

In order to ensure $H_{p_{\text{enc}}}(Z)$ is fixed, we model the conditional density $p(z|x)$ with a deep neural network that outputs the means $\mu = F(x)$ of a spherical Gaussian with fixed variance $\sigma = 1$. We assume that the generating distribution is the same as with GANs in the previous section. The objectives for training a bi-directional adversarial model then becomes:

$$\arg \min_{D} \mathbb{E}_{p_{\text{enc}}}[\log D(x, z)] + \mathbb{E}_{p_{\text{dec}}}[\log(1 - D(x, z))]$$
$$\arg \max_{F, G} \mathbb{E}_{p_{\text{enc}}}[\log(1 - D(x, z))] + \mathbb{E}_{p_{\text{dec}}}[\log D(x, z)] + \beta I_{p_{\text{enc}}}(X, Z). \tag{23}$$

We will show that a bi-directional model trained in this way has the benefits of higher mutual information, including better mode coverage and reconstructions.

**Experiment: bi-directional adversarial model with mutual information maximization** In this section we compare MINE to existing bi-directional adversarial models in terms of euclidean reconstructions, reconstruction accuracy, and MS-SSIM metric (Wang et al., 2004). One of the potential features of a good generative model is how close the reconstructions are to the original in pixel space. Adding MINE to a bi-directional adversarial model gets us closer to this objective. We train MINE on datasets of increasing order of complexity: a toy dataset composed of 25-Gaussians, MNIST, and the CelebA dataset.

Figure 5 shows the reconstruction ability of MINE compared to ALI. Although ALICE does perfect reconstruction (which is in its explicit formulation), we observe significant mode-dropping in the sample space. MINE does a balanced job of reconstructing along with capturing all the modes of the underlying data distribution.

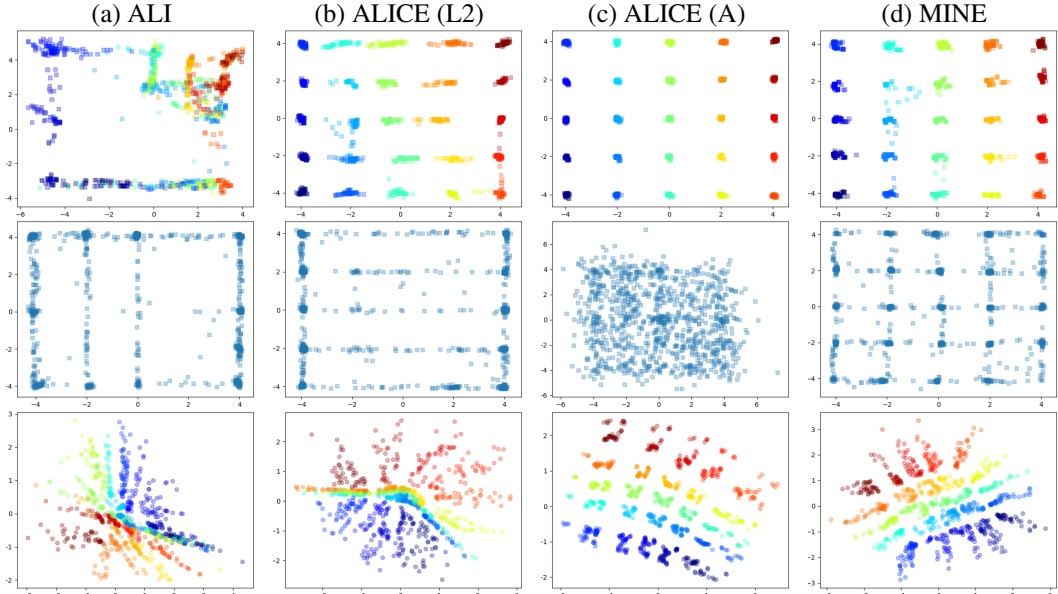

Figure 5: Reconstructions, samples, and embeddings from adversarially learned inference (ALI) and variations intended to increase the mutual information. Shown left to right are the baseline (ALI), ALICE with the L2 loss to minimize the reconstruction error, ALI with an additional adversarial loss, and MINE. Top to bottom are the reconstructions, samples from the prior, and the embeddings. ALICE with the adversarial loss has the best reconstruction, though at the expense of sample quality. Overall, MINE provides both very good reconstructions and the best mode representation in its samples.

Next, we use MS-SSIM (Wang et al., 2004) scores to measure the likelihood of generated samples within the class. Table 1 compares MINEto the existing baselines in terms of euclidean reconstruction errors, reconstruction accuracy, and MS-SSIM metric. MINE does a better job than ALI in terms of reconstruction errors by a good margin and is competitive to ALICE with respect to reconstruction accuracy and MS-SSIM. Table 2 shows that MINE's effect on reconstructions is even more dramatic when compared to ALI and ALICE. Thus showing that MINE can efficiently operate in a truly large scale setting.

| MNIST | | | |
|---|---|---|---|
| | Reconstruction Error(Euclidean) | Reconstruction Accuracy(%) | MS-SSIM |
| ALI | 14.24 | 45.95 | 0.97 |
| ALICE | 5.20 | 98.17 | 0.98 |
| MINE | 9.73 | 96.10 | 0.99 |

Table 1: Comparison of MINE with other bi-directional adversarial models in terms of euclidean reconstruction error, reconstruction accuracy, and ms-ssim on MNIST dataset. We used MLP both in the generator and discriminator identical to the setting described in Salimans et al. (2016) and MLP Statistics network for this task. MINE does a decent job compared to ALI in terms of reconstructions. Though the explicit reconstruction based baselines do better than MINE in terms of tasks related to reconstructions, they lag behind in MS-SSIM scores.

## 4.4 INFORMATION BOTTLENECK

The Information Bottleneck (IB, Tishby et al., 2000) is an information theoretic method for extracting relevant information, or yielding a representation, that an input $X \in \mathcal{X}$ contains about an output $Y \in \mathcal{Y}$. An optimal representation of $X$ would capture the relevant factors and compress $X$ by diminishing the irrelevant parts which do not contribute to the prediction of $Y$. IB was recently

| CelebA | | | |
|---|---|---|---|
| | Reconstruction Error(Euclidean) | Reconstruction Accuracy(%) | MS-SSIM |
| ALI | 53.75 | 57.49 | 0.81 |
| ALICE | 92.56 | 48.95 | 0.51 |
| MINE | 36.11 | 76.08 | 0.99 |

Table 2: Comparison of MINE with other bi-directional adversarial models in terms of euclidean reconstruction error, reconstruction accuracy, and MS-SSIM on CelebA faces dataset. We can see that the trend remains same from MNIST results. MINE achieves a substantial decrease in reconstruction errors without compromising on better MS-SSIM score.

covered in the context of deep learning (Tishby & Zaslavsky, 2015). As such, IB can be seen as a process to construct an approximate of minimally sufficient statistics of the data. IB seeks a feature map, or encoder, $q(Z \mid X)$, that would induce the Markovian structure $X \rightarrow Z \rightarrow Y$. This is done by minimizing the IB Lagrangian,

$$\mathcal{L}[q(Z \mid X)] = H(Y|Z) + \beta I(X, Z) \tag{24}$$

which appears as a the standard cross-entropy loss augmented with a regularizer promoting minimality of the representation (Achille & Soatto, 2017). Here we propose to estimate the regularizer with MINE.

**Related works and information bottleneck with MINE**   In the discrete setting, Tishby et al. (2000) uses the Blahut-Arimoto Algorithm Arimoto (1972), which can be understood as cyclical coordinate ascent in function spaces. While the information bottleneck is successful and popular in a discrete setting, its application to the continuous setting was stifled by the intractability of the continuous mutual information. Nonetheless, the Information Bottleneck was applied in the case of jointly Gaussian random variables in Chechik et al. (2005).

In order to overcome the intractability of $I(X; Z)$ in the continuous setting, Alemi et al. (2016); Kolchinsky et al. (2017); Chalk et al. (2016) exploit the variational bound of (Barber & Agakov, 2003) to approximate the conditional entropy in $I(X; Z)$. The approaches of the aforementioned works differ only on their treatment of the marginal distribution of the bottleneck variable. Alemi et al. (2016) assumes a standard multivariate normal marginal distribution, Chalk et al. (2016) uses a Student-t distribution, and Kolchinsky et al. (2017) uses non-parametric estimators. Due to their reliance on a variational approximation, all the method above require a tractable density for the approximate posterior.

MINE estimate the mutual information directly. As such, it allows for general posterior as it does not require densities. Thus MINE allows the use of general encoders/posteriors.

**Experiment: Permutation-invariant MNIST classification**   Here, we demonstrate an implementation of the Information Bottleneck objective on a permutation invariant MNIST using MINE. We use a similar setup as Alemi et al. (2016), except that we do not use their approach to averaging the weights. The architecture of the encoder is an MLP with two hidden layers and an output of 256 dimensions. The decoder is a simple softmax. As Alemi et al. (2016) is using a variational bound on the conditional entropy, their approach requires a tractable density. They opt for a conditional Gaussian encoder $z = \mu(x) + \sigma \odot \epsilon$, where $\epsilon \sim \mathcal{N}(0, I)$. As MINE does not require a tractable density, we consider three type of encoders:

- A Gaussian encoder as in Alemi et al. (2016)

- An *additive noise encoder*, $z = enc(x + \sigma \odot \epsilon)$

- A *propagated noise encoder*, $z = enc([x, \epsilon])$.

Our results can be seen in Table 3, and this shows MINE as being superior in all of these settings.

| Variational Bottleneck | Misclassification rate(%) |
|---|---|
| Variational Bottleneck | 1.37% |
| MINE (Gaussian) | 1.26% |
| MINE(Propagated) | 1.24% |
| MINE(Additive) | 1.19% |

Table 3: Permutation Invariant MNIST misclassification rate using information bottleneck methods.

## 5 CONCLUSION

We proposed a mutual information estimator, which we called the mutual information neural estimator (MINE), that is scalable in dimension and sample-size. We demonstrated the efficiency of this estimator by applying it in a number of settings. First, a term of mutual information can be introduced alleviate mode-dropping issue in generative adversarial networks (GANs, Goodfellow et al., 2014). Mutual information can also be used to improve inference and reconstructions in adversarially-learned inference (ALI, Dumoulin et al., 2016). Finally, we showed that our estimator allows for tractable application of Information bottleneck methods (Tishby et al., 2000) in a continuous setting.

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

# 6 APPENDIX

## 6.1 ON THE CONNECTION BETWEEN MUTUAL INFORMATION AND STRUCTURE

We argue that mutual information can be also understood as a measure of the amount of 'structure' between two sets of random variables. It remains to define what structure is. In a loose sense, structure can be seen as a realization of the manifold hypothesis. If we consider the observables as the coefficients of a coordinate system, then we can say that there is strong structure if that coordinate system can be expressed as a function of a smaller set of coordinates. In this sense, structure implies dependency. Baring the difficult case of causality, the existence of structure reveals itself empirically

through co-occurrence. We illustrate this perspective by considering distributions on natural image manifolds.

Consider a random image in $[0,1]^d$ by randomly sampling the intensity of each pixel independently. This image will show very little structure when compared to an image sampled form the manifold of natual images, $\mathcal{M}_{nature} \subset [0,1]^d$, as the latter is is bound to respect a number of physically possible priors (such as smoothness). We expect the mutual information of the pixels of images arising from $\mathcal{M}_{nature}$ to be high. Differently put, the larger the number of simultaneously co-occurring subset of pixels in $[0,1]^d$, the higher the mutual information. In the language of cumulants tensors, the larger ponderation of higher order cumulants tensor in the cumulant generating function of the joint distribution over the pixels, the higher the mutual information, and the more structure there is to be found amongst the pixels. Note that the case of mutually independent pixels corresponds to joint distribution where the only cumulants contributing the joint distribution are of order one. This is the corner case where the joint distribution equals the product of marginals. Thus in order to assess the amount of structure it is enough to score how the joint distribution is different from the product of marginals. As we show in the paper, this principle can be extended to different divergences as well.

## 6.2 CONSISTENCY PROOFS

This section presents the proofs of the Lemma leading to the consistency result in Theorem 2. In what follows, we will assume that the input space $\Omega = \mathcal{X} \times \mathcal{Z}$ is a compact domain of $\mathbb{R}^d$ and all measures are absolutely continuous with respect to the Lebesgue measure. To avoid unnecessary heavy notation, we denote $\mathbb{P} = \mathbb{P}_{XZ}$ and $\mathbb{Q} = \mathbb{P}_X \otimes \mathbb{P}_Z$ for the joint distribution and product of marginals. We will restrict to families of feedforward functions with continuous activations, with a single output neuron, so that a given architecture defines a continuous mapping $(\omega, \theta) \to T_\theta(\omega)$ from $\Omega \times \Theta$ to $\mathbb{R}$.

### 6.2.1 PROOF OF LEMMA 1

Consider the function $T^* = \log \frac{d\mathbb{P}}{d\mathbb{Q}}$ that saturates the bound (11). By construction, $T^*$ satisfies:

$$\mathbb{E}_\mathbb{P}[T^*] = I(X, Z), \qquad \mathbb{E}_\mathbb{Q}[e^{T^*}] = 1 \tag{25}$$

For a function $T$, the (positive) gap $I(X, Z) - \hat{I}(T)$ can be written as

$$I(X, Z) - \hat{I}(T) = \mathbb{E}_\mathbb{P}[T^* - T] + \log \mathbb{E}_\mathbb{Q}[e^T] \leq \mathbb{E}_\mathbb{P}[T^* - T] + \mathbb{E}_\mathbb{Q}[e^T - e^{T^*}] \tag{26}$$

where we used the inequality $\log x \leq x - 1$.

Fix $\eta > 0$. We first consider the case where $T^*$ is *bounded* from above by a constant $M$. By the universal approximation theorem (see corollary 2.2 of Hornik (1989)[5]), we may choose a feedforward network function $T_{\hat\theta} \leq M$ such that

$$\mathbb{E}_\mathbb{P}|T^* - T_{\hat\theta}| \leq \frac{\eta}{2} \quad \text{and} \quad \mathbb{E}_\mathbb{Q}|T^* - T_{\hat\theta}| \leq \frac{\eta}{2} e^{-M} \tag{27}$$

Since $\exp$ is Lipschitz continuous with constant $e^M$ on $(-\infty, M]$, we have

$$\mathbb{E}_\mathbb{Q}|e^{T^*} - e^{T_{\hat\theta}}| \leq e^M \mathbb{E}_\mathbb{Q}|T^* - T_{\hat\theta}| \leq \frac{\eta}{2} \tag{28}$$

From Equ 26 and the triangular inequality, we obtain:

$$|I(X, Z) - \hat{I}(T_{\hat\theta})| \leq \mathbb{E}_\mathbb{P}|T^* - T_{\hat\theta}| + \mathbb{E}_\mathbb{Q}|e^{T^*} - e^{T_{\hat\theta}}| \leq \frac{\eta}{2} + \frac{\eta}{2} \leq \eta \tag{29}$$

In the general case, the idea is to partition $\Omega$ in two subset $\{T^* > M\}$ and $\{T^* \leq M\}$ for a suitably chosen large value of $M$. Given $S \subset \Omega$, we will denote by $\mathbb{1}_S$ its characteristic function, $\mathbb{1}_S(\omega) = 1$ if $\omega \in S$ and 0 otherwise. $T^*$ is integrable with respect to $\mathbb{P}$[6], and $e^{T^*}$ is integrable with

---

[5]Specifically, the argument relies on the density of feedforward network functions in the space $L^1(\Omega, \mu)$ of integrable functions with respect the measure $\mu = \mathbb{P} + \mathbb{Q}$.

[6]This can be seen from the identity (Györfi & van der Meulen, 1987)

$$\mathbb{E}_\mathbb{P}\left|\log \frac{d\mathbb{P}}{d\mathbb{Q}}\right| \leq D_{KL}(\mathbb{P} \,||\, \mathbb{Q}) + 4\sqrt{D_{KL}(\mathbb{P} \,||\, \mathbb{Q})}$$

respect to $\mathbb{Q}$, so by the dominated convergence theorem, we may choose $M$ so that the expectations $\mathbb{E}_{\mathbb{P}}[T^* \mathbb{1}_{T^*>M}]$ and $\mathbb{E}_{\mathbb{Q}}[e^{T^*} \mathbb{1}_{T^*>M}]$ are lower than $\eta/4$. Just like above, we then use the universal approximation theorem to find a feed forward network function $T_{\hat{\theta}}$, which we can assume without loss of generality to be upper-bounded by $M$, such that

$$\mathbb{E}_{\mathbb{P}}|T^* - T_{\hat{\theta}}| \leq \frac{\eta}{2} \quad \text{and} \quad \mathbb{E}_{\mathbb{Q}}|T^* - T_{\hat{\theta}}| \mathbb{1}_{T^* \leq M} \leq \frac{\eta}{4} e^{-M} \tag{30}$$

We then write

$$\begin{aligned}
\mathbb{E}_{\mathbb{Q}}[e^{T^*} - e^{T_{\hat{\theta}}}] &= \mathbb{E}_{\mathbb{Q}}[(e^{T^*} - e^{T_{\hat{\theta}}}) \mathbb{1}_{T^* \leq M}] + \mathbb{E}_{\mathbb{Q}}[(e^{T^*} - e^{T_{\hat{\theta}}}) \mathbb{1}_{T^* > M}] \\
&\leq e^M \mathbb{E}_{\mathbb{Q}}[|T^* - T_{\hat{\theta}}| \mathbb{1}_{T^* \leq M}] + \mathbb{E}_{\mathbb{Q}}[e^{T^*} \mathbb{1}_{T^* > M}] \\
&\leq \frac{\eta}{4} + \frac{\eta}{4} = \frac{\eta}{2}
\end{aligned} \tag{31}$$

where the inequality in the second line arises from the convexity and positivity $\exp$. Equations 30 and 31, together with the triangular inequality, lead to Equation 29, which proves the Lemma.

### 6.2.2 PROOF OF LEMMA 2

Let $\mathcal{F}$ be the family of functions $T_\theta \colon \Omega \to \mathbb{R}$ defined by a given network architecture, with a compact parameter space $\Theta \subset \mathbb{R}^d$. For a given collection of $n$ *iid* samples, MINE is constructed by means of the empirical measures, denoted here $\mathbb{P}_n, \mathbb{Q}_n$ for simplicity. The defining equation 13 and the triangular inequality give

$$|\widehat{I(X;Z)}_n - \sup_{T_\theta \in \mathcal{F}} \hat{I}(T_\theta)| \leq \sup_{T_\theta \in \mathcal{F}} |\mathbb{E}_{\mathbb{P}}[T_\theta] - \mathbb{E}_{\mathbb{P}_n}[T_\theta]| + \sup_{T_\theta \in \mathcal{F}} |\log \mathbb{E}_{\mathbb{Q}}[e^{T_\theta}] - \log \mathbb{E}_{\mathbb{Q}_n}[e^{T_\theta}]| \tag{32}$$

The continuous function $(\theta, \omega) \to T_\theta(\omega)$, defined on the compact domain $\Theta \times \Omega$, is bounded. So the functions $T_\theta$ are uniformly bounded by a constant $M$, i.e $|T_\theta| \leq M$ for all $\theta \in \Theta$. Since $\log$ is Lipschitz continuous with constant $e^M$ in the interval $[e^{-M}, e^M]$, we have

$$|\log \mathbb{E}_{\mathbb{Q}}[e^{T_\theta}] - \log \mathbb{E}_{\mathbb{Q}_n}[e^{T_\theta}]| \leq e^M |\mathbb{E}_{\mathbb{Q}}[e^{T_\theta}] - \mathbb{E}_{\mathbb{Q}_n}[e^{T_\theta}]| \tag{33}$$

Since $\Theta$ is compact and the feedforward network functions are continuous, the families of functions $T_\theta$ and $e^{T_\theta}$ satisfy the uniform law of large numbers (Van de Geer, 2000). Given $\eta > 0$ we can thus choose $N \in \mathbb{N}$ such that $\forall n \geq N$ and with probability one,

$$\sup_{T_\theta \in \mathcal{F}} |\mathbb{E}_{\mathbb{P}}[T_\theta] - \mathbb{E}_{\mathbb{P}_n}[T_\theta]| \leq \frac{\eta}{2} \quad \text{and} \quad \sup_{T_\theta \in \mathcal{F}} |\mathbb{E}_{\mathbb{Q}}[e^{T_\theta}] - \mathbb{E}_{\mathbb{Q}_n}[e^{T_\theta}]| \leq \frac{\eta}{2} e^{-M} \tag{34}$$

Together with Equations 32 and 33, this leads to

$$|\widehat{I(X;Z)}_n - \sup_{T_\theta \in \mathcal{F}} \hat{I}(T_\theta)| \leq \frac{\eta}{2} + \frac{\eta}{2} = \eta \tag{35}$$

### 6.3 ADAPTIVE CLIPPING

Here we assume we are in the context of GANs described in Sections 4.2 and 4.3, where the mutual information shows up as a regularizer in the generator objective.

Notice that the generator is updated by two gradients. The first gradient is that of the generator's loss, $\mathcal{L}_{gen}$ with respect to the generator's parameters $\Theta_{gen}$, $g_{unsup} := \frac{\partial \mathcal{L}_{gen}}{\Theta_{gen}}$. The second flows the the mutual information estimate to the generator, $g_{MI} := -\frac{\partial \widehat{I(X;Z)}}{\partial \Theta_{gen}}$. If left unchecked, because mutual information is unbounded, the latter can overwhelm the former, leading to a failure mode of the algorithm where the generator puts all of its attention on maximizing the mutual information and ignores the adversarial game with the discriminator. We propose to adaptive clip the gradient from the mutual information so that its Frobenius norm is at most that of the gradient from the discriminator. More formally, we have

$$g_{adapted} := \min(\|g_{unsup}\|, \|g_{MI}\|) \frac{g_{MI}}{\|g_{MI}\|}. \tag{36}$$

Note that adaptive clipping can be considered in any situation where MINE is to be maximized.

### 6.4 BOUND ON THE RECONSTRUCTION ERROR

Here we clarify relationship between reconstruction error and mutual information, by proving the bound in Equ 22. We begin with a definition:

**Definition 6.1** (Reconstruction Error). We consider encoder and decoder models giving conditional distributions $p_{\mathrm{enc}}(z|x)$ and $p_{\mathrm{dec}}(x|z)$ over the data and latent variables. If $p(x)$ denotes the marginal data distribution, the reconstruction error is defined as

$$\mathcal{R} = \mathbb{E}_{x\sim p(x)}\mathbb{E}_{z\sim p_{\mathrm{enc}}(z|x)}[-\log p_{\mathrm{dec}}(x|z)] \tag{37}$$

We can rewrite the reconstruction error in terms of the joints $p_{\mathrm{enc}}(x,z) = p_{\mathrm{enc}}(z|x)p(x)$ and $p_{\mathrm{dec}}(x,z) = p_{\mathrm{dec}}(x|z)p(z)$. Elementary manipulations give:

$$\mathcal{R} = \mathbb{E}_{(x,z)\sim p_{\mathrm{enc}}}\log\frac{p_{\mathrm{enc}}(x,z)}{p_{\mathrm{dec}}(x,z)} - \mathbb{E}_{(x,z)\sim p_{\mathrm{enc}}}\log p_{\mathrm{enc}}(x,z) + \mathbb{E}_{z\sim p_{\mathrm{enc}}(z)}\log p(z) \tag{38}$$

where $p_{\mathrm{enc}}(z)$ is the marginal on $\mathcal{Z}$ induced by the encoder. The first term is the KL-divergence $D_{KL}(p_{\mathrm{enc}} \parallel p_{\mathrm{dec}})$ ; the second term is the joint entropy $H_{p_{\mathrm{enc}}}(x,z)$. The third term can be written as

$$\mathbb{E}_{z\sim p_{\mathrm{enc}}(z)}\log p(z) = -D_{KL}(p_{\mathrm{enc}}(z) \parallel p(z)) - H_{p_{\mathrm{enc}}}(z)$$

Finally, the identity

$$H_{p_{\mathrm{enc}}}(x,z) - H_{p_{\mathrm{enc}}}(z) := H_{p_{\mathrm{enc}}}(z|x) = H_{p_{\mathrm{enc}}}(z) - I_{p_{\mathrm{enc}}}(x,z) \tag{39}$$

yields the following expression for the reconstruction error:

$$\mathcal{R} = D_{KL}(p_{\mathrm{enc}}(x,z) \parallel p_{\mathrm{dec}}(x,z)) - D_{KL}(p_{\mathrm{enc}}(z) \parallel p(z)) - I_{p_{\mathrm{enc}}}(x,z) + H_{p_{\mathrm{enc}}}(z) \tag{40}$$

Since the KL-divergence is positive, we obtain the bound:

$$\mathcal{R} \leq D_{KL}(p_{\mathrm{enc}}(x,z) \parallel p_{\mathrm{dec}}(x,z)) - I_{p_{\mathrm{enc}}}(x,z) + H_{p_{\mathrm{enc}}}(z) \tag{41}$$

which is tight whenever the induced marginal $p_{\mathrm{enc}}(z)$ matches the prior distribution $p(z)$.

