# OpenReview forum: "MINE: Mutual Information Neural Estimation"
_ICLR.cc/2018/Conference — Reject_

### Official Review · AnonReviewer3 · 2017-11-24
**Potentially workable idea, but needs a lot more work**

**Rating:** 3
**Confidence:** 4

**Review:**

Summary
======================================================================
The authors propose an estimator for the Shannon Mutual Information that is based on
the Donsker Varadhan lower bound. The idea is to choose an expressive class of functions
(in this case, parametrized by a NN) and maximise a statistic. The authors present
some applications for the proposed estimator.

While the idea of using the Donsker-Varadhan lower bound is interesting and potentially
workable, the theory is not strong and the experiments are far from compelling to warrant
acceptance.

Detailed Review
======================================================================

Why is this called an online MI estimator? Nothing about the formalism or the estimator
uses an online learning approach.

Despite the authors' claims, the estimator does not come with any theoretical guarantees.
- What is your definition of strongly consistent? Since the MI is a scalar quantity,
  strong consistency is the same as (weak) consistency.
- The proof is not given, and I don't think the proposed estimator would be consistent if
  you use a fixed class of networks T. There is a necessarily a bias between the class of
  functions the network can approximate and all bounded functions.

Missing citations: There is a ton of recent work on estimating mutual information
that the authors have missed. These are a few but you should also look at papers that
cite these and are cited by these.
- Kandasamy et al 2015. Nonparametric von Mises estimators for entropies, divergences and
  mutual informations.
- Singh & Poczos 2016. Finite-sample analysis of fixed kNN density functional estimators.
- Moon et al 2017. Ensemble estimation of Mutual Information.

In Algorithm 1, why do the samples have to be inside the loop? What is wrong with
applying the last two lines on the same dataset? On the same note, do you really need
\bar{z}^(i) to be different from z^(i)?

The authors claims in the introduction that the non-parametric methods make critical
assumptions while GANs do not is misleading. Many of the methods make assumptions for the
theoretical analysis - in practice, some, if not most of them work well even when the
assumptions do not hold. Similarly, if you want to prove something about GANs you
probably have to make assumptions too.

Experiments:
The authors present 4 use cases. All of them are toy settings and none of them make a
compelling case for the proposed estimator.
- In the GAN setting, I am failing to see why one would use a MI regularizer over an
  entropic regularizer. It seems like what you need is entropy, and it is not clear what
  happens to the conditional entropy term when you maximize MI.
- Section 4.3: The bound on the reconstruction error is dropping a KL(p(z)||q(z)) term
  and the authors don't really discuss how lose this is.
- The authors make claims about scalability with n and d but none of the experiments
  show the evaluation times compared to simpler estimators.

Minor
- I thought there were several unnecessary tangential discussions that didn't really add
  much to the paper. For instance, Section 3.3 was unnecessary given that all the
  experiments solely focused on the Shannon case. The para after theorem 1 on the
  compression lemma doesn't add much. Even the definitions of the Shannon MI and Theorem
  1 could have been stated without appealing to measure theory constructs.
- Figure 1: This is perhaps not the cleanest way to present this graph. Perhaps consider
  plotting the error in a log-scale might work better.

---

> ### Author Response · Authors · 2018-01-05
> **Answer to AnonReviewer3**
>
> We thank the reviewer for her/his comments and feedback.
>
> 1) To avoid confusion with the terminology, motivated by the reviewer's comment,  we changed the name of the estimator from "Online Mutual Information Estimator" to "Mutual Information Neural Estimator (MINE)".
>
> 2) The theoretical analysis of the estimator has been substantially improved, clarified and strengthened in the new version.   The updated version of the paper now includes a precise definition of (strong) consistency (Section 3.2, Definition 3.2), a precise statement for our consistency  result (Section 3.2, Lemma 1 and 2 Theorem 2), together with a full proof (Appendix 6.2).
>
> Regarding the specific concerns expressed by the reviewer:
>     - MI is a scalar function of N random samples from the  data distribution (as such it is a random variable).  Consistency is concerned with convergence towards the true estimated quantity as N goes to infinity: weak consistency corresponds to convergence in probability, while strong consistency corresponds to almost sure  convergence. This is very similar to the standard framework of M-estimators. We refer to the Def 3.2 for a precise and workable definition of what we mean by consistency.
>     - Working with a *fixed* network indeed induces an approximation bias. One of the points in the proof is that this approximation bias is controlled and can be set to be arbitrarily small by the universal approximation theorem (see Lemma 1).  In the proof added in the new version, we tried to show in a clear way how
> the consistency problem separates into an approximation problem (dealt in Lemma 1) and an estimation problem (dealt in Lemma 2). We hope that with the material added, the definition, result and proof are now crystal clear.
>
> 3) Following the reviewer's suggestion, we have added recent references to
> the related work section. We emphasize however that a lot of  work on MI estimation in the literature use non-parametric methods, which tend to suffer from the curse of dimensionality. One of the main purpose of MINE is  to address this problem by proposing a highly scalable MI estimator by means of neural networks.
>
> 4) In Algorithm 1, In algorithm 1 (x^(i), z^(i)) are samples from the joint; and (x^(i), \bar{z}^(i)) are samples from the product of marginals.  We have added in Section 3.1 a quick explanation for how to get samples of the product of marginals.
>
> 5) We thank the reviewer for the comment on implicit assumptions in GANS. We have updated our paper accordingly.
>
> 6) Regarding the comment on the toy experiments, we kindly point the reviewer to section 4.3 where we use our estimator to improve reconstructions in Bi-directional generative models on MNIST and CelebA as well as to section 4.4 where we apply our estimator to information bottleneck based regularization on MNIST.
>
> 7) We clarified our presentation of the framework in Section 4.2. The key point of our proposal is to use the mutual information as proxy for the (generally intractable) entropy. In fact, optimizing the mutual information with respect to the generator parameters is equivalent to optimizing the entropy. This can be very easily seen in the case of a discrete z distribution: if the generator G(z) is a deterministic function of z, the conditional entropy vanishes and the mutual information reduces to the entropy. In the general case,  note that -- both in theory and in practice --, to avoid divergence issues, (e.g Gaussian) noise is typically added to the generator so as to regularize  the deterministic conditional G(z) |z to a  well-defined density (it amounts to working with a Gaussian approximation of the Dirac distribution). Now, the conditional entropy H(G(z)| z) depends only on the variance and is independent of the  parameters. So the mutual information and entropy differ by a constant and have the same gradient.
>
> 8) For the reconstruction error bound, we use the positivity of the Kullblack-Leibler divergence in Equation 41 to drop it thus transforming the equality in an inequality. Please note that this inequality is tight since at convergence p_enc(z) and p(z) are matched. Following the reviewer comment we spelled out in more detail the reconstruction bound derivation in Appendix 6.4.
>
> 9) Regarding scalability, let us emphasize that by `scalability' we mean that our estimator does not suffer as much from the curse of dimensionality and did not mean to refer to evaluation times.
>
> 10)  Following the reviewer comment, we have removed the discussion on the compression lemma. More generally we worked on improving and re-balancing the presentation in the updated version of the paper.
>
> 11) Following the reviewer comment, we have improved the presentation in Figure 1.

---

### Official Review · AnonReviewer2 · 2017-11-27
**Apply existing variational representations of f-divergences to build a MI estimator mostly for GANs**

**Rating:** 5
**Confidence:** 4

**Review:**

The authors present an estimator for the mutual information (MI) based on the Donsker-Varadhan representation for the KL divergence and its generalization to arbitrary f-divergences by Ruderman et al. While that last work introduced an estimator based on optimization over the unit ball in an RKHS, the current work propose to use a parametric function class given by a neural network (I'd suggest that the authors make this point more explicit, as currently it's not totally clear what their actual contribution is and how their work compares to the prior art they cite). The authors show that such an estimator can be used to train models with less mode-dropping in adversarial models.

The work is quite straightforward, but improves over similar work in the GAN space by Nowozin et al. by using Ruderman's tighter variational representation instead of Nguyen's one.

The paper contains many typos and grammatical errors and the authors should do an exhaustive proof-reading. More problematic is that, right after eq. 10, the authors mention "We show in the Appendix that OMIE has the desirable strong consistency and convergence properties". However, the appendix doesn't contain such a proof. Is it missing from the submitted version? I don't think that such a consistency proof is strictly necessary for a paper like this, but for the review to be accurate I need to see the proof. Since I can't find it, I assume it does not exist. In that case, the authors should give less emphasis to the MI estimator itself and more to the empirical properties and applications.

The authors present some experiments comparing different estimators of MI applied to synthetic data. Figure 1 is hard to read, I suggest the authors try to come up with a more legible plot. Figure 2 is also a bit surprising, why show error for 50 dimensions but estimates for 2 dimensions? Since these experiments are quick to run, it would be helpful to get more information on how the gap between the methods change as the dimensionality increases (e.g. a surface plot with d and # of iterations on the x and y axes). Also it would be highly beneficial to compare with the method in Ruderman at al., so that people interested in MI estimation but who don't plan on using the estimator as part of a neural net architecture can get some idea on how the inductive bias of NNs compare to RKHS.

In the caption to Fig. 3 the authors state "The OMIEGAN generator learns a distribution with a high amount of structured noise", which I find hard to understand. Probably the authors can be a bit more precise than saying "structured noise".

I would recommend dropping the Information Bottleneck section to focus on showing more convincingly the impact of OMIE in GANs. The experiments section currently looks rushed and lacking in depth.

In summary, this work provides value by introducing a (previously known) superior f-divergence variational representation to the GAN community. The mode-collapse prevention via MI maximisation is also interesting and deserves more experimental attention to make the paper stronger.

---

> ### Author Response · Authors · 2018-01-05
> **Answer to AnonReviewer2**
>
> We thank the reviewer for her/his feedback.
>
> 1) The updated version has been exhaustively proof-read; typos have been fixed, presentation and notations have also been substantially improved.
>
> 2) The consistency result was indeed claimed with no proof in the first submitted version. The new version now contains the precise consistency result (Section 3.2, Theorem 2) with a full proof (Appendix 6.2).
> More generally, the whole theoretical part of the paper has been strengthened.
>
> 3)  We have updated and improved Figures 1 and 2 according to the reviewer's recommendations.
>
> 4) We agree that it would be very interesting to compare MINE with the estimation of Nguyen et al or Ruderman et al  using RKHS methods. We've left this for future work, to focus on the main applications in Section 4. However we emphasize that one of the main purpose of our approach is to provide a highly scalable estimator; and we do expect MINE to have much better scalability properties than non-parametric methods such as RKHS.

---

### Official Review · AnonReviewer1 · 2017-11-27
**A proposed new method to estimate mutual information which requires more thorough experiments.**

**Rating:** 5
**Confidence:** 4

**Review:**

This paper presents a new method for estimation of mutual information (MI) based on the Donsker-Varhan (DV) representation of KL-divergence. This representation requires the calculation of a supremum over a set of functions and a lower bound is achieved when a neural network is used for the maximisation of it. Computing the DV representation also requires evaluating expectations wrt to the distributions of interest, the proposed method uses Monte-Carlo estimates based on the empirical distributions.

The experiments evaluating the quality of the OMIE estimator for mutual information should be more thorough to make a point that OMIE beats competing estimators. The bivariate Gaussian case presented in Figure 1 is not a very relevant test case as estimating MI is especially difficult in higher dimensions. It would also be interesting to know the number of samples used as the ratio nbr dimensions/samples matters for estimation quality. The caption for Figure 2 mentions “bivariate Gaussians of dimension 50”, do the author mean two Gaussians of dimension 50 each?

The results of the proposed method on the swiss-roll dataset look good, however the authors only provide a comparison to a classic GAN where it seems more natural to compare with the other works on mode-dropping for GAN cited in the related works section. A comparison with InfoGAN and Dai et al. would be especially relevant to evaluate the effectiveness of OMIE.

On the application of OMIE to the Information Bottleneck (IB) problem:
How was the optimization of the objective exactly performed? How are gradients calculated? Is the reparametrisation trick used? More details should be provided on the results presented in table 3. Are the results obtained on the test set? What was the value of beta and to which values of I(X,Z) and I(Z,Y) does it correspond? Was the misclassification rate averaged over multiple runs?

The generalization to f-divergences is interesting but seems rather straightforward.

The second line of equation (20) does not make sense to me, it is not equivalent to the first line.

The methods proposed in Alemi et al. and Chalk et al. differ also in the way the bounds are estimated, not only in the choice of the marginal distribution.

The authors mention that strong consistency and convergence properties (page 3) are proven in the appendix, however I could not find them.

---

> ### Author Response · Authors · 2018-01-05
> **Answer to AnonReviewer1**
>
> We thank the reviewer for her/his  comments and insights.
>
> 1) Non-parametric methods to estimate the mutual information tend to suffer from the curse of dimensionality and do not scale. The raison d'être of MINE is to address this point.
> The main goal of the comparative experiments in Section 4.1 is to illustrate, in a setting where the mutual information is tractable,  that MINE performs as well as non-parametric methods, and outperforms parametric competitors (already guaranteed theoretically in the case of Nguyen's divergence estimation, as emphasized at the end of Section 2.2).  The results of the next sections, which focus on GANs and information bottleneck methods, also indirectly illustrate the efficacy of the estimator.
>
> In the caption of Fig 2, we indeed meant 2 Gaussians of dimension 50 each.
>
> 2) Applying MINE to other set-ups would be a good test for MINE and is definitely a line of research to pursue. We emphasize however that our set-up in Section 4.2 was designed specifically to address mode dropping in Gans, which was not the case of the mentioned related  works such as InfoGan. To the best of our knowledge, addressing mode dropping by means of mutual information estimation is done here for the first time.
>
> 3) Precisions concerning Information bottleneck experiments:
> To keep comparison sensible we followed the same set-up of Alemi 2016. We used Adam with a learning rate of 1e-4. The beta parameter in the IB equation was set to 1e-3. We did not need to use the reparametrization trick as our method provides a backpropable estimate of I(X;Z). The results are obtained on the test set. The misclassification rate was averaged over 10 runs.
>
> 4) We only briefly mention in Section 3.3 the possibility extend the construction to f-divergence using the approach of Ruderman et al.  We believe it was worth mentioning, as it opens new avenue of investigations.
>
> 5) We improved the presentation and notation of the Information bottleneck Section.
>
> 6) We thank the reviewer for the precision about the work of Alemi et al. and Chalk et al. We have updated our paper accordingly.
>
> 7) The consistency result was indeed claimed with no proof in the first submitted version. The new version now contains the precise consistency result (Section 3.2, Theorem 2) with a full proof (Appendix 6.2).

---

### Author Response · Authors · 2018-01-05
**Changes from the original submission**

We updated version contains several clarifying changes, including substantial improvement of the theoretical analysis, improved presentation and notations, added references, and better plots.
We humbly ask that the reviewers take a fresh look at the paper as it now stands.

The main modifications include the following:

1) We changed the name of the estimator, from "Online Mutual Information Estimator (OMIE)" to "Mutual information Neural Estimator (MINE)". We changed the paper's title  accordingly.

2) The most substantial changes concern the theoretical part, Section 2 (background), Section 3 and the Appendix:
     * The presentation of the Donsker-Varadhan bound has been improved, a very simple proof has been added, as well as a comparison with the f-divergence bound of Nguyen et al.
     * The Section 3 and the Appendix 6.2 now include a theorem with a full proof of the consistency of the estimator.
     * We also improved and clarified the derivation of the bound of the reconstruction error in Appendix 6.4.

3) We improved the presentation and made some notational changes in Section 4.3  and 4.4 to improve clarity.

4) We improved the plots in Figure 1, by splitting them into two groups of estimators to ease readability and comparison.

5) We clarified and added references in the related work paragraphs throughout Section 4.

---

### Decision · Program_Chairs · 2018-01-29
**ICLR 2018 Conference Acceptance Decision**

**Decision:**

Reject

**Comment:**

The main ideas of the paper are promising, but there remain several important concerns.  The initial submission contained significant typos and missing proofs.  In the revision, these have been added, but the main article before references and appendices now stands at 11 pages.  It is therefore lacking focus and readability required for presentation in this venue.

From AnonReviewer2: "The work is quite straightforward, but improves over similar work in the GAN space by Nowozin et al. by using Ruderman's tighter variational representation instead of Nguyen's one."  This comment raises concerns about the overall novelty of the proposed approach.

AnonReviewer1 raised concerns about the suitability of the chosen experimental setup, namely two Gaussians.  It is not clear that the current experiments provide a sufficiently detailed evaluation in the context of a highly competitive venue like ICLR.  Given that the theoretical contribution was incomplete in the initial submission and has not been fully verified, empirical evaluation has a higher importance in the assessment of this paper.